# Passivated Impedimetric Sensors for Immobilization-Free Pathogen Detection by Isothermal Amplification and Melt Curve Analysis

**DOI:** 10.3390/bios12050261

**Published:** 2022-04-20

**Authors:** Matthias Steinmaßl, Jamila Boudaden, Güven Edgü, Lena Julie Freund, Simon Meyer, Noa Mordehay, Melissa Soto, Hanns-Erik Endres, Jost Muth, Dirk Prüfer, Wilfried Lerch, Christoph Kutter

**Affiliations:** 1Fraunhofer Research Institution for Microsystems and Solid State Technology EMFT, 80686 Munich, Germany; matthias.steinmassl@emft.fraunhofer.de (M.S.); sensorics@he-endres.de (H.-E.E.); wilfried.lerch@emft.fraunhofer.de (W.L.); christoph.kutter@emft.fraunhofer.de (C.K.); 2Physics Institute, Universität der Bundeswehr München, 85577 Neubiberg, Germany; 3Fraunhofer Institute for Molecular Biology and Applied Ecology IME, 52074 Aachen, Germany; gueven.edgue@ime.fraunhofer.de (G.E.); Lena.Julie.Freund@ime.fraunhofer.de (L.J.F.); jost.muth@ime.fraunhofer.de (J.M.); 4Hochschule München, University of Applied Sciences, 80686 Munich, Germany; m.meyer.simon@mail.de (S.M.); noa.mordehay@gmail.com (N.M.); 5Center for Systems Biotechnology, Fraunhofer Chile Research Foundation, Santiago 7500588, Chile; melissa.soto.garces@gmail.com

**Keywords:** point-of-care, impedance, virus detection, LAMP, melt curve analysis, microheating

## Abstract

The ongoing SARS-CoV-2 pandemic demonstrates that the capacity of centralized clinical diagnosis laboratories represents a significant limiting factor in the global fight against the newly emerged virus. Scaling up these capacities also requires simple and robust methods for virus diagnosis to be easily driven by untrained personnel in a point-of-care (POC) environment. The use of impedance sensors reduces the complexity and costs of diagnostic instruments and increases automation of diagnosis processes. We present an impedance point-of-care system (IMP-POCS) that uses interdigitated electrodes surrounded by an integrated heating meander to monitor loop-mediated isothermal amplification (LAMP) and melt curve analysis (MCA) consecutively in a short time. MCA permits distinguishing false- from true-positive results and significantly raises the validity of pathogen detection. Conclusively, the herein-developed miniaturized total analysis system (µTAS) represents a powerful and promising tool for providing reliable, low-cost alternatives to standard clinical diagnosis.

## 1. Introduction

Since early 2020, the newly emerged severe acute respiratory syndrome coronavirus 2 (SARS-CoV-2) has dominated the globalized world, pushing our healthcare systems to the limit. Current detection for SARS-CoV-2 infection is routinely performed by specific nucleic acid amplification tests (NAATs), using specialized equipment and trained personnel. In this context, the method of reverse transcription-quantitative polymerase chain reaction (RT-qPCRs) is considered the “gold standard”. For this purpose, total RNA is purified from respiratory samples of potentially infected individuals and subsequently subjected to RT-qPCR for viral RNA detection. Here, RT and PCR can either be performed consecutively within a single tube and buffer (one-step assay), or sequentially in separate vessels with optimal buffers for each reaction (two-step assays). Ideally, 3 to 4 h are required until the diagnosis is finalized.

In the current pandemic, significant limitations in standard diagnostics have emerged worldwide, e.g., in the supply of sufficient quantities of appropriate nasopharyngeal swabs, RNA extraction kits, or RT-PCR detection reagents.

To expand testing capacity, several new compact cartridge-based point-of-care (POC) analyzers, originally developed for other indications such as meningitis, have been adapted for SARS-CoV-2 detection [1].

The main advantage of these space-saving cartridge analyzers is that they combine and automate nucleic acid extraction, purification, amplification, and detection in a (mostly) self-explanatory RT-qPCR-like device [2]. Some platforms such as the Cobas SARS-CoV-2 8800 system (Roche Molecular Diagnostics, Pleasanton, CA, USA) even allow for high-throughput automated screenings with a potential capacity of 1056 tests in 8 h [3]. However, many of these new test devices are based on disposable cartridge systems designed for single use. The fabrication of these cartridges is complex and cost-intensive, as it requires for example functionalization of sensor surfaces or particles or the use of expensive labels [4,5]. Current market leaders such as Abbott ID NOW or Ceiphas GeneXpert use fluorescent dyes that are activated by incorporation into double-stranded DNA (dsDNA) [6,7]. However, as the acquisition and material costs of these devices are high, the corresponding POC tests can only be offered by a few laboratories. In addition, these POCs (costs approx. EUR 200) are currently used mainly at airports, where quick results are needed [8]. Consequently, expensive equipment remains a major bottleneck of current testing and mass screening strategies outside laboratory environments in general.

To overcome these limitations, other powerful methods of nucleic acid sequence-specific tests, which do not require temperature cycling, have been developed in recent decades. Referred to as isothermal amplification methods, these techniques vary in their basic concept and have different merits and drawbacks. Their common features are rapid reaction, operation at constant temperature, and independence of bulky, elaborate, and costly equipment. For example, innovative experimental detection approaches derived from the combination of clustered regularly interspaced short palindromic repeats (CRISPR) and recombinase polymerase amplification (RPA) that are usually conducted at constant temperature, ranging between 37 °C and 42 °C, were just recently introduced [9]. Assays of *Specific High-sensitivity Reporter unLOCKing* (SHERLOCK) [10] and *One-tube RT-RPA- DNA Endonuclease-Targeted CRISPR Trans Reporter* (OR-DETECTR) using Cas13 and Cas12a enzyme, respectively [11], seem to represent promising and future-orientated approaches for pathogen detection. Nonetheless, in this special case, novelty can be seen as a disadvantage, as there is currently no accurate data of mass validation and field trials.

Another PCR alternative that became popular as a NAAT outside laboratories is known as loop-mediated isothermal amplification (LAMP). LAMP reactions are highly specific, and the detection limit is similar to that of a standard PCR due to the involvement of 4–6 specific primers targeting 6–8 distinct nucleic acid target regions. Unlike standard PCR reactions, LAMPs are robust and tolerant of inhibitors allowing analysis of crude or minimally processed input samples, thereby providing convenient and fast assay setups [12,13]. At a constant temperature of 65 °C, supplied by any heat source such as a water bath, this “hyper-priming” on the target sequence driven by a strand-displacing DNA polymerase takes place and leads to a high-speed exponential amplification in less than 30 min [14]. Subsequently, the amplification products can be visualized by a variety of methods, including turbidity detection, real-time fluorescence detection (when used with LAMP fluorescent dye), and pH-based colorimetric detection [15]. Besides that, other physical effects are known that can be exploited to monitor pathogen detection techniques without the use of labeling dyes or immobilization. The bulk electrical properties of suspended nucleic acids were investigated by Ma et al. [16] using impedimetric sensors with regard to fragment length, concentration, and denaturation. They characterized a droplet of DNA solution with a two-electrode configuration without any immobilization on the electrode surface and showed that the sensing electrode can distinguish between single-stranded (ss-) and double-stranded DNA (dsDNA). Even earlier, a digital polymerase chain reaction system was presented [17] that consists of a microfluidic setup and interdigitated electrodes to perform PCR and detect the amplification products by electrical impedance measurement. The same method was used to perform real-time measurement of PCR inside a laboratory thermal cycler [18]. Additionally, the denaturation of dsDNA can also be measured by electrical impedance spectroscopy. It was reported that the execution of melt curve analysis on impedance shows comparable results to experiments conducted on a qPCR cycler [19,20]. Unlike previously described methods for PCR monitoring, immobilized DNA strands on the electrode surface were necessary to perform melt curve analysis.

In this context, we present an approach that provides rapid and efficient detection of SARS-CoV-2 by melt curve analysis (MCA) based on virus-specific LAMP products. Both LAMP and MCA are performed consecutively on passivated impedimetric sensors. Thus, together with a temperature control module, a promising miniaturized total analysis system (µTAS) for successful real-time application and impedance-based monitoring of LAMP and MCA was developed (see Figure 1). Our approach opens up new opportunities for innovative, cost-effective, and future-orientated NAAT-POCTs.

## 2. Materials and Methods

### 2.1. SARS-CoV-2 N Gene RNA Reference

#### 2.1.1. PCR

PCRs were performed according to our earlier published protocol [21]. Different dilutions of commercial plasmid 2019-nCoV_N_Positive Control (Integrated DNA Technologies, Coralville, IA, USA) were used as a template (100,000–100 plasmid copies), together with T7-promoter containing primer T7-N1F (5′-TAATACGACTCACTATAG-GGGACCCCAAAATCAGCGAAAT-3′) and the NotI-N2 reverse primer (5′-TATCATGACGGCGGCCGCGCGCGACATTCCGAAGAA-3′) to amplify sufficient amounts of DNA covering a stretch of 944 bases of the viral N gene (coordinates: bases from 28,287 to 29,230 of the Wuhan wildtype, GenBank ID: MN908947.3 [22]). Amplification products were separated by 1% agarose gel electrophoresis, purified using the NucleoSpin Gel and PCR Clean-up kit according to the provider’s instructions (Macherey-Nagel, Düren, Germany), pooled, and the resulting DNA yield was determined by spectrophotometry using a Nanodrop 2000 device (Thermo Fisher Scientific, Waltham, MA, USA).

#### 2.1.2. In Vitro Transcription and RNA Isolation

Subsequent in vitro transcription (IVT) was carried out with 2 µg of the purified N gene amplicon using the TranscriptAid T7 High Yield Transcription Kit (Thermo Fisher Scientific, Waltham, MA, USA) according to the manufacturer’s instructions. IVT products were purified using the Monarch Total RNA Miniprep Kit (NEB, Ipswich, MA, USA) according to the manufacturer’s protocol. Optional DNase treatment was included and prolonged to 25 min. The concentration of the resulting RNA was determined by spectrophotometry.

### 2.2. Accredited SARS-CoV-2 Reference Samples

In the Federal Republic of Germany, the INSTAND society has been appointed by the German Medical Association as a reference institution for external quality assessment (EQA). The versatile interlaboratory comparison program from INSTAND is accredited in all representative specialist disciplines of laboratory diagnostics by the German Accreditation Body (DAkkS) according to DIN EN ISO/IEC 17043:2010. Here, two crude heat-inactivated cell-culture supernatants from SARS-CoV-2 INSTAND reference samples were used along with mentioned IVT N gene RNA as an alternative for clinical patient samples that were not available at the time of this study.

In advanced impedimetric LAMP experiments, the EQA sample 417010 containing a 1:2500 diluted thermally inactivated cell-culture supernatant derived from BetaCoV/Passau/ChVir21652/2020 (VOC B.1.1.7) was used as a reference for SARS-CoV-2-positive specimens. As a negative control, EQA cell lysate sample 409040 was used. For this SARS-CoV-2-negative sample, participants in EQA testing reported positive or questionable results [23].

### 2.3. Mini-LAMP Assay

A WarmStart Colorimetric LAMP 2X LAMP kit with UDG and dUTP included in the reaction master mix (E1804, NEB, Ipswich, MA, USA) was used for all experiments presented herein. The oligonucleotide set for the detection of SARS-CoV-2 (Table 1) includes F3/B3, FIP/BIP, and FL/BL loop primers. The FIP and FL components further incorporate 5′-biotine and FAM-labels, respectively, to enable additional lateral-flow dipstick (LFD) verification using the AMODIA DetectLine Basic kit (AMODIA Bioservice, Braunschweig, Germany) [21,24]. Primers were ordered from Integrated DNA Technologies (Leuven, Belgium).

### 2.4. Procedures

Prior to the experiments, chemicals were allowed to adapt to room temperature for 45 min. After cleaning the sensors in an ultrasonic bath with 70% ethanol inside the reaction chambers, the sensors were rinsed with deionized (DI) water and dried with N_2_. LAMPs were carried out according to the manufacturer’s protocol, except that 40 mM guanidine hydrochloride was added. The reaction mix was filled into the reaction chambers and incubated at room temperature for 5 min. Subsequently, a temperature of 65 °C was applied by the sensor-controlled heating ability (see Section 3). The colorimetric mix was used to validate the success of the amplification by optical inspection, as the included phenol red indicator initially shows pink (pH 8.2–8.6) and should turn yellow during a successful LAMP reaction. This switch of color is initiated when nucleotides are incorporated into the 3’ end of nascent DNA products by releasing protons, which results in a remarkable pH drop (below pH 7) [15].

Melt curve analysis (MCA) was carried out by executing impedance measurements at temperatures from 65 to 100 °C in steps of 0.25 °C. The acquisition of one impedance measurement took around 15 s; therefore, the speed of temperature change was 1 °C/min. The theoretical melting temperature *T_m_* of the SARS-CoV-2 amplicon was calculated with the online tool uMelt [25] using the Unified SL model.

For laboratory measurements, an impedance analyzer (HP 4192A), together with Analog Devices ADG1207 multiplexers, was used to perform the impedance measurements, as well as to validate the measurements made by our µTAS. The calculation of the capacitance was performed by assuming an equivalent parallel R-C circuit using the equation.
(1)$$C=-\frac{1}{2\pi fZ\sin\theta }$$

An overview of this experimental laboratory setup is provided in Figure 2. Additionally, to the described temperature control system and the impedance measurement, it shows the reaction chambers, which are realized by gluing a half-cut PCR tube strip to the impedance sensor and the corresponding cap strips for a hermetic enclosure. On the top of the caps, an aluminum plate was placed as a heating block to prevent any condensation of sample solution on the caps.

### 2.5. Fabrication of the Packaged Sensors

The impedance sensors were fabricated inside a class 1000 clean room. A 5 nm TiW adhesion layer and 175 nm gold were deposited on quartz glass substrates (150 mm wafer). The metal was structured by a lithographic etching process and then tempered. A 2 µm thick polyimide layer was deposited on the top of the structured metal as a passivation layer. The sensors were packaged in a sensor-on-hole configuration on a flexible printed circuit board (FLEX-PCB) [26]. The developed flexible packaging provides both electrical connectivity and hermetic encapsulation of the sensor. To provide reaction vessels, a PCR tube strip was cut in half, the upper part was glued onto the hole of the FLEX-PCB. PCR tube strips are usually made of polypropylene, which has unfortunately poor adhesion properties. Pretreatment with Loctite SF 770 primer solution and gluing with Loctite 424 ethyl-based instant adhesive produce acceptable hermetic sealing properties for at least three consecutive experiments. In initial iterative experiments, we managed to reduce the required reaction volume of the sample solution from 50 µL initially to 35 µL (data not presented in this paper).

## 3. Results and Discussion

### 3.1. Newly Developed Sensors

Impedance sensors with different geometries were fabricated (Figure 3). From the center to the outside, a sensor consists of a reference electrode (dark color), two comb-like electrodes, a thermistor, and a heating meander. Only the reference electrode, the bondpads, and a sawing line on the chip edge are not covered by the polyimide layer (indicated by the darker color in Figure 3). The distance between the electrode fingers varied between 6, 10, and 50 µm. The resistance of the thermistor was used as a measure of the temperature. A current was applied to the heating meander to increase the temperature of the sensor. The impedance was measured between the two comb-like electrodes. The reference electrode, in the center, was implemented just in case the sensors will be used for other applications in the future.

COMSOL Multiphysics was used to calculate the capacitance for the different geometries in air and in contact with various liquid media. The model consists of a multilayered stack having a glass substrate, electrodes, a passivation layer, and the medium above this passivation. Figure 4a shows the electric potential in a cross-sectional view between two neighboring electrode fingers in water. The white lines represent the streamlines of the electric field, and their thickness corresponds to the magnitude. It can be seen that the majority of the electric field energy is accumulated inside the substrate and the passivation layer, as their dielectric constant is small compared to water (glass = 4.2, polyimide = 3.6, and water = 80). However, the capacitance of the sensors is still influenced by the dielectric properties of the above media, especially if the thickness of the passivation is smaller than the electrode finger gap. In this case, the passivation layer was 2 µm thick. By assuming *ϵ* = 100 for 1% DNA solution [16], the full-scale change (FSC) for different geometries was calculated (see Figure 4b). Effectively, the wider the electrode gap, the higher the FSC, but the lower the capacitance, which makes it harder for the readout electronics to measure such small changes in the capacitance. Our microcontroller-based readout method measures a capacitance change of 10 fF. Nevertheless, from previous studies, we know that a base capacitance value of >10 pF is required for a stable measurement [27].

### 3.2. Description and Validation of the Temperature Control System

The designed system to conduct the experiments is shown in Figure 5. The system is powered by a 12 V source (e.g., in a car’s cigarette lighter). It can handle and measure eight impedance sensors simultaneously. Figure 5b shows a FLEX-PCB with eight reaction vessels connected to the system. For each channel, there is a temperature measurement provided by the thermistor of the impedance sensor (R*_temp_* in Figure 5a). Together with R*_ref_*, it builds a resistive voltage divider, where the voltage is dependent on the resistance of the thermistor and therefore the temperature of the sensor’s surface. An 8-channel 16-bit SAR ADC (Texas Instruments ADS8332) compares this voltage to the 2 V reference voltage of a linear voltage regulator and transfers the reading to the microcontroller (STM32L476RG) via the SPI protocol. The microcontroller features eight timer controllers used to generate a PWM timer signal for each channel. The PWM signal, together with a MOSFET, adjusts the current through the heating meander of the impedance sensor. The temperature measurement was calibrated by placing the system in an incubator and measuring the ADC reading for each channel at different temperatures of 25, 40, 55, and 70 °C. A PID algorithm was used to calculate the duty cycle for the PWM and therefore the heating current from the ADC reading to control the temperature of the sensor’s surface. Prior to the experiments, the calibration data were loaded onto the microcontroller, and a Python framework [27] was executed to perform instrument control and data acquisition from a PC.

The microcontroller features a touch-sensing controller that has been used as a measurement engine for these types of sensors in previous projects. It utilizes the charge-transfer method for the measurement of the sensor capacitance. Only one external component (*C_ref_* in Figure 5a) is needed [27,28].

The temperature control concept was tested by recording the ADC readings via the serial interface from the microcontroller (see Figure 6). The temperature set point started at 75 °C and was increased by 0.5 °C every 5 s. The response time of the control is lower than the data acquisition interval of 100 ms. The temperature stability is good and around 0.2 °C.

A heatable aluminum plate was placed over the caps to prevent the condensation of liquid on the caps of the tube strip. During the experiment, the temperature of this plate was controlled to the same set value. A 35 µL volume of DI water was added to the reaction vessels, heated, and kept at a constant temperature of 65 °C. After two hours, 35 µL was still present inside the vessel. The loss due to evaporation can be neglected, and a hermetically sealed vessel can be assumed. These results are encouraging because errors induced by any evaporation effect of the LAMP mix on the impedance could be ruled out.

### 3.3. LAMP Reaction

To select the optimum measurement parameters, first, a frequency spectrum of the impedance data was acquired in the range from 10 to 400 kHz with a sinusoidal voltage stimulus of 0.3 V amplitude. As the impedance sensor can be modeled as a parallel RC circuit, for high frequencies, the capacitive contribution to the impedance is predominant. It was chosen as the dielectric constant was supposed to be linked to the DNA configuration. The measurement was performed with 400 kHz as the maximum frequency. This represents the highest possible measurement frequency of the touch-sensing controller integrated into the microcontroller, used to develop a point-of-care (POC) demonstrator. The experiment was carried out in parallel on impedance sensors with different electrode distances. Figure 7 shows the spectra for the 6 µm sensor (Figure 7a), 10 µm sensor (Figure 7b), and 50 µm sensor (Figure 7c). The color indicates the temporal evolution from 0 min (brown) to 35 min (green) in steps of 5 min. The buckling of the data near 30 kHz results from a switch in the current measurement range by the impedance analyzer. Overall, the impedance decreases with time, but the maximum impedance change is observed for the 50 µm sensor. A purely capacitive behavior is represented by a straight line in the logarithmic impedance plot and a phase angle of −90°. This is solely the case for frequencies higher than 50 kHz with the 50 µm sensor (see Figure 7c). For smaller gap values, there is a negative resistive part of the impedance.

In this case, the heating meander acts as an additional electrode. As the voltage at the heating meander (5 V) is higher compared to the stimulating AC voltage of the impedance measurement (0.3 V), the current from the heater to the electrode decreases with increasing AC voltage. Current transport is provided by the polyimide passivation. As reported in the literature [29], wet polyimide causes leakage currents in the presence of a high concentration of ions. The increased resistive part of the impedance at lower frequencies is also caused by this leakage current. Especially for smaller distances of the electrode fingers and for lower frequencies, the conductivity of the wet polyimide contributes to the impedance. By analyzing the Nyquist plots of the measurements in Figure 8, it can be seen that the impedance change can be mainly reasoned by a change in the imaginary part of the impedance, which is represented by the capacitance of a parallel RC circuit.

The idea behind the sensing concept is to look at the change in the dielectric properties, which can be extracted from the measured capacitance of the passivated sensor. These results show that at sufficiently high frequencies, this concept is valid. Capacitance values were calculated from these results and are shown with the time-dependent change of impedance and phase in Figure 9. for both frequencies of 25 and 400 kHz. The deployed impedance analyzer exhibits a fairly high error of 1% and an acceptable error of 0.4% for 25 kHz and 400 kHz, respectively. Effectively, the impedance analyzer measures the current response from a voltage stimulus. The lower the frequency, the higher the impedance, the lower the current, and the lower the measurement accuracy. Therefore, 400 kHz was chosen as the optimum frequency for both the impedance measurement in the laboratory and for the microcontroller-based capacitance measurement used in the miniaturized total analysis system (µTAS). Previous reports have shown that impedimetric sensors with 10 pF capacitance can be measured with good accuracy using a touch-sensing controller [30]. Although the results at 25 kHz show a higher full-scale change (FSC) for all measured quantities, the higher frequency was chosen for accurate capacitance measurement. The capacitance part of the impedance is below 1 pF, which is hard to measure with a high-precision impedance analyzer, but even harder with a microcontroller-based readout.

During the LAMP reaction, an increase in capacitance was observed, and the measurement curve is similar to characteristic curves from qPCR measurements [21]. After an initial equilibrium phase with a stable measurement value, the exponential growth starts (in this case, after 10 min) and reaches saturation after a short time. Presumably, during the LAMP reaction, a change in the dielectric constant occurs as single dNTPs get incorporated into the nascent cDNA. In theory, the presence of dsDNA leads to an increase in the dielectric constant [31]. The dissolved DNA is a negatively charged polyanion, and counterions are accumulated on the negative charge positions. As these counterions can travel along the backbone of the DNA molecule, it is induced to form a strong dipole in an electric field. The dipoles of the DNA base pairs do not contribute to the dielectric properties of the medium, as they are complementary and cancel each other.

The capacitive full-scale change of the LAMP reactions surpasses the expectations derived from the simulated values. With more reasonable assumptions, the simulation could be improved by specifying the dielectric constant in more detail. The initial LAMP Master Mix has a lower dielectric constant due to the replacement of H_2_O molecules by other ingredients.

The high ohmic proportion of the impedance can be reasoned by the use of the ADG1207 multiplexers, which exhibit an extremely low charge injection of 0.5 pC but a high on-resistance of up to 300 Ω. Regarding the selection of the sensor’s geometry, for the following experiments, the 6 µm sensor was used, as it showed the highest capacitance value. Compared to the 50 µm sensor, which shows an FSC of around 20%, the 6 µm sensor reaches an FSC of 9% (compare Figure 9f), which still delivers a good signal-to-noise ratio for these measurements.

To prove the capability of this measurement to quantify the nucleic acid target from the LAMP monitoring, LAMP reactions with different sample concentrations of 160, 16, and 4.6 pg/µL were performed on these sensors. Figure 10a shows that for lower concentrations, the reactions start to reach exponential growth at a later point in time. A statistical analysis of all LAMP experiments conducted on capacitive sensors is shown in Figure 10b. The full-scale change is located between 1% and 5% for the 6 µm sensor and 17% for the 50 µm sensor. All experiments shown in this graph were performed with a virus concentration of 160 pg/µL, and the x-axis shows the time when the exponential increase of the signals starts to be recorded.

### 3.4. Melt Curve Analysis

After the LAMP reaction was completed, the heating control was turned off. Once room temperature was reached, the remaining sample solution was checked by pipetting, and the color change of the colorimetric buffer was noted. In the case of a positive reaction and a negligible volume loss, MCA was then performed by closing the caps, placing the aluminum heating block, and then heating the sensor from 65 to 100 °C in steps of 0.25 °C. At the end of each step, the capacitance of the sensors was measured. After this ramp, heating was turned off, and after cooling down to room temperature, the remaining reaction volume was repeatedly checked. For the data analysis, the results were smoothed by the Savitzky–Golay filter method with 25 data points and plotted against the temperature. The negative first derivative was calculated by OriginLab data analysis software.

As a reference, MCA was compared to parallel-conducted experiments, and results were obtained using a qPCR cycler (Figure 11). On the qPCR cycler, the SARS-CoV-2 amplicon showed a *T_m_* of 88.25 °C, which is comparable to the value predicted by uMELT (88.25 °C), while the nontemplate control displayed no *T_m_* value. Analogously performed impedimetric measurement results are shown in Figure 12.

Three MCA experiments were performed simultaneously on three different 6 µm impedance sensors using the SARS-CoV-2 LAMP amplicon sample. Figure 12a shows the capacitance change for the three equivalent sensors compared to a non-template control (NTC). As the sensors exhibit different base capacitance, the data are shown by the change in capacitance referred to as the capacitance value at 70 °C. Since data progress between 70 and 100 °C is similar, good reproducibility is proven. As expected, MCA of the nontemplate control exhibited no detectable value at all, as no amplification was observed, while all investigated amplicons of the SARS-CoV-2 RNA template were detected correctly. Thus, the method can discriminate between positive and negative samples.

As can be seen in Figure 12b, which displays the mean value of the negative derivative of the three curves with the standard deviation (error bars), there is an initial decrease in capacitance at 85 °C that can be traced back to the denaturation of the amplicon, and an increase in capacitance at 90 °C is observed. The maximum decrease in capacitance is at 88.25 °C, while the maximum increase in capacitance is at 92.5 °C.

Both a decrease at 88.25 °C and an increase at 92.5 °C were observed for the MCA with the SARS-CoV-2 amplicons for all three experiments. However, the latter extreme value was not visible in the MCA analogously performed on a qPCR cycler (see Figure 11). There are several aspects during the denaturation of cDNA that contribute to a change in the dielectric constant of the sample solution. An increase in capacitance can be explained, e.g., by the breakup of nucleic base pairs. This leads to an additional dipole contribution by the nucleic acid bases. A decrease in capacitance could be reasoned by the fact that single-stranded nucleic acid has a smaller polarization moment than double-stranded one. The different points of temperature might be related to the necessity of a reorientation of the strand. However, the dipole properties of nucleic acids are also dependent on the distribution of the four different nuclei bases, which have a huge impact on the geometric orientation of the strands. Simulations of these folding mechanisms might be required to fully understand the capacitive measurements during MCA. Additionally, impedance investigation of MCA on other RNA sequence amplicons might also reveal the underlying mechanisms of the different processes. However, the absence of any capacitance changes in the melt curve of nontemplate controls shows that the main idea of distinguishing between positive and false-positive amplicons is possible by capacitive measurement.

During denaturation, the cDNA, which is generated by the LAMP reaction, is separated into ssDNA. The denatured ssDNA does not contribute to the total dielectric constant of the medium in the same portion as dsDNA. Therefore, a melt curve should be characterized by a decrease in the impedance, which is in line with previous studies [19,20,32]. In these earlier studies on impedimetric MCA, immobilization of DNA strands onto the surface of an impedance sensor was necessary to identify the *T_m_* value [19,20]. It is worth underlining that in this work, melt curve analysis was conducted without any requirement for functionalization/immobilization of RNA or DNA on sensor surfaces or particles. The impedimetric sensors enable obtaining with good precision the *T_m_* of the suspended dsDNA in the LAMP product.

### 3.5. Transfer to a Microcontroller-Based Readout System

Figure 13 shows the development of a POC demonstrator based on an impedimetric sensor for the virus detection using the combined LAMP and MCA processes. To facilitate the use of the system by the operators, the prototype was extended by a Raspberry-Pi-powered user guidance interface.

The display guides the user through the necessary steps to be performed, starting with the preparation of the mixture, the heating, and the readout of the LAMP reaction results first and MCA reaction results afterwards.

SARS-CoV-2 RT-LAMPs and subsequent MCAs were repeated using this improved demonstrator. The 6 µm sensor was used, as it exhibits a capacitance higher than 10 pF at the TSC measurement frequency. For this experiment, the EQA sample 417010 containing an Alpha VOC cell-culture specimen was used as a positive control (PC) for SARS-CoV-2. The virus-free cell lysate sample 409040 that demonstrated false-positive results in multiple gold-standard RT-PCR assays, as reported by INSTAND, was used as a negative control (NC). Both crude samples were diluted 1/10 before using them as templates in impedimetric and colorimetric RT-LAMP, followed by lateral-flow dipstick analysis (LAMP-LFD assays). Figure 14 shows the results of the LAMP reaction (Figure 14a) and subsequent MCA (Figure 14b). The touch-sensing controller (TSC) of the used STM32L476 is sensitive enough to measure the process of the LAMP reaction and monitor the capacitive change versus time. The accuracy is lower compared to the impedance analyzer results, but the outcome is comparable. The full-scale change (FSC) is at 3%. Data of the MCA were smoothed using a Savitzky–Golay filter. The negative derivation of the smoothed data reveals a maximum loss in capacitance at 88.25°C for the PC sample, which is consistent with the obtained melting temperature from earlier qPCR cycler control experiments (see Figure 11 and Figure 12), as well as theoretically predicted *T_m_* by uMELT. The capacitance increase obtained from the impedance analyzer measurement cannot be resolved by the TSC measurement. For the EQA 409040 NTC (red line in Figure 14b), no capacitance change was observed. As the primer set used (Table 1) includes Biotin- and FAM-labeled primer components, LFD analysis was additionally carried out at the end and successfully demonstrated the correctness of our LAMP and MCA impedimetric measurements (Figure 14c).

## 4. Conclusions and Outlook

A two-stage impedimetric POCT system for the detection of pathogens was successfully developed (IMP-POCS) and demonstrated on pandemic-causing SARS-CoV-2 and the descendant Alpha variant of concern. During the first stage of this novel approach, pathogenic target nucleic acid is detected by monitoring a (reverse transcription-, RT-) LAMP process with target-specific primers immobilization free on the impedance sensor. The resulting measurement provides a (RT-) LAMP reaction readout and endpoint detection in real-time (e.g., Figure 9). The results are similar to those accomplished with professional laboratory thermal cyclers used for classical qPCRs. Importantly, there is no actual need for intensive upstream sample preparation (e.g., RNA extraction). In the second step, the LAMP product is then analyzed by MCA (see Figure 12). As different temperatures (up to 95 °C) can be run, practically any kind of other potential future POCTs is enabled, for example, promising CRISPR/Cas-RPA assays that require other, lower permanent thermal conditions, e.g., 37 °C [9]. Thus, the safety advantage of an “unopened” one-vessel, one-step test such as for gold-standard clinical analysis and diagnosis is maintained. Any potential incorrect LAMP amplicons, such as self-amplified primer conglomerates, etc. [33,34], can be distinguished from correct (e.g., viral) target-induced specific LAMP products by MCA [35]. The specificity of the LAMP amplification, as verified by the impedimetric results of the MCA, was conclusively confirmed by lateral-flow dipstick (LFD) chromatography. With this increase in the validity of LAMP reactions, the presented concept delivers a robust and cost-effective method for future NAAT-POCTs. It is plausible that this method could also be used for the quantification of the virus load (see Figure 12a). Of note, this new combined LAMP and MCA impedance sensor, in contrast to cartridge analyzers, neither requires the complicated functionalization of electrode surfaces [4], nor the use of fluorescent dye labeling or other nanomaterial particles, thereby saving costs and labor time in general.

Compared to existing impedance studies on amplification or denaturation processes [16,17,18,19,20], this research does not rely on electrical contact between the sample solution and the bare metal of the sensing electrodes. As the sensing electrodes are not directly in contact with the sample, it might be a promising solution to make the passivation layer interchangeable. The impedance sensor in that case could be reused and therefore costs per test would be decreased. In the end, this first experimental “out-of-the-box” demonstrator prototype for impedimetric LAMP assays managed to successfully “mimic” real-time qPCR and MCA results for SARS-CoV-2 variant-resistant detection that are usually achieved in professional laboratories.

## Figures and Tables

**Figure 1 biosensors-12-00261-f001:**
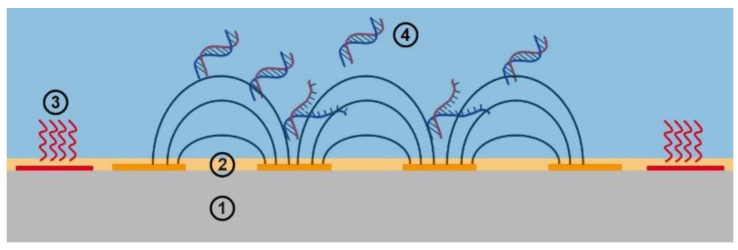
Schematic drawing of the virus detection concept. The sensing element is an impedance sensor ➀ that consists of passivated interdigitated electrodes ➁, which are surrounded by a thermistor and heating elements ➂. The capacitance measured by the impedance sensor is influenced by the dielectric properties of the sample solution. The idea is to sense the configuration and concentration of suspended nucleic acids ➃ on the surface of the impedance sensor.

**Figure 2 biosensors-12-00261-f002:**
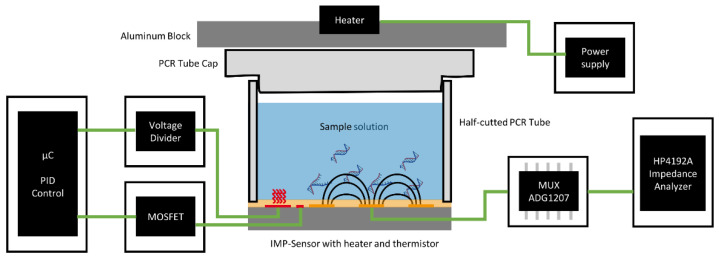
Overview of the experimental laboratory setup. The temperature control system was used to set the desired temperature for the sample solution. An impedance analyzer was used to measure the impedance of the sensor via a multiplexer IC ADG1207. A heated aluminum block prevents condensation of the sample solution on the PCR tube caps.

**Figure 3 biosensors-12-00261-f003:**
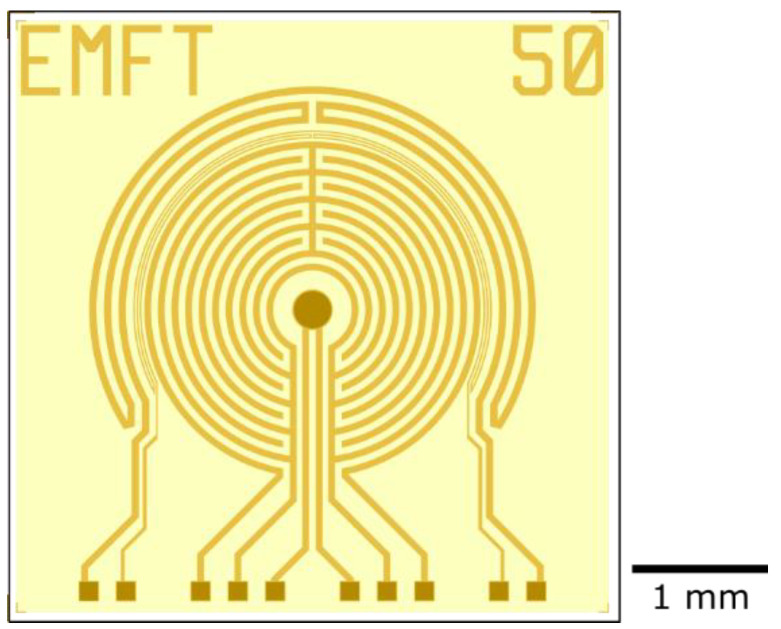
Impedance sensor. From the center to the outside of the chip, the structures in the gold layer are visualized: a reference electrode, two interdigitated electrodes, a temperature resistor, and a heating meander. The darker color visualizes areas that are not covered by the polyimide passivation (yellow), such as the reference electrode and the bondpads.

**Figure 4 biosensors-12-00261-f004:**
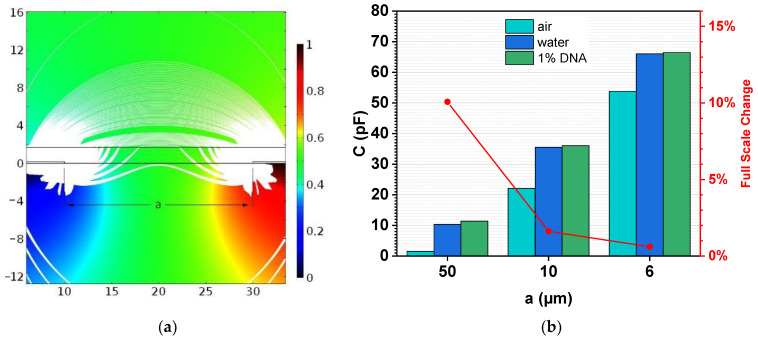
(**a**) Visualization of the electric potential and the streamlines of the electric field (white) in water. The graph shows a cross-section through the chip between two electrode fingers. One is grounded (blue) and the other one at a potential of 1 V (red). (**b**) The simulated values for the capacitance of the three geometry variants are shown. From the capacitance in water and 1% DNA solution, the expected full-scale change was estimated.

**Figure 5 biosensors-12-00261-f005:**
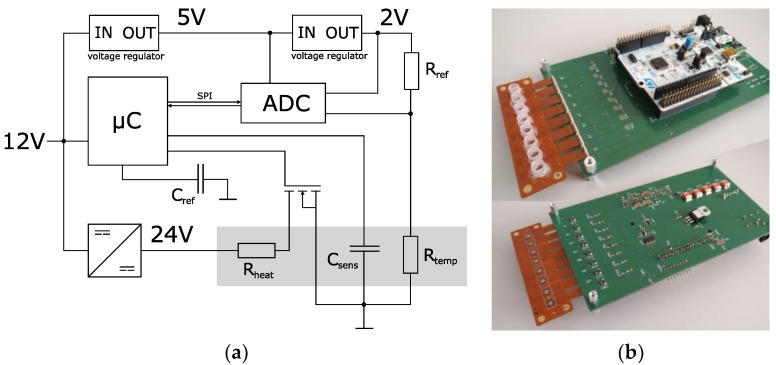
Temperature control system. (**a**) Block diagram. The temperature control is realized by three components. Temperature measurement is performed by ADC with the resistive voltage divider between R*_ref_* and the thermistor R*_temp_*. Heating current calculation is performed by the microcontroller (µC) with a PID. Current through the heating meander R*_heat_* is controlled by the Timer PWM and the MOSFET. A UART interface enables instrument control and data acquisition through a PC. (**b**) Photograph. The orange FLEX-PCB incorporates 8 impedance sensors in a sensor-on-hole packaging. A half-cut PCR tube strip is glued onto the PCB. The white PCB is the Nucleo evaluation board of the STM32L476RG.

**Figure 6 biosensors-12-00261-f006:**
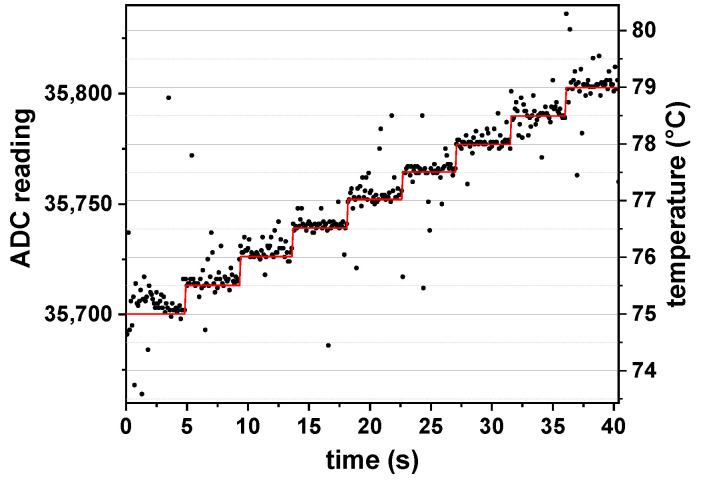
Recorded ADC readings from the temperature control system (Figure 5) for a given temperature set point (red). The data were recorded every 100 ms. A 0.5 °C change needs a response time of less than 100 ms. Despite single measurement deviations, the fast PID update timer stabilizes the temperature control inside an interval of 0.1 °C.

**Figure 7 biosensors-12-00261-f007:**
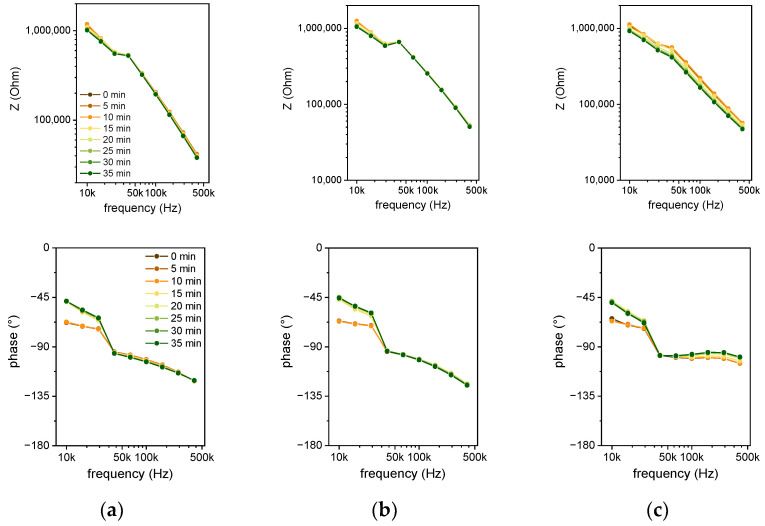
Impedance spectra of (**a**) the 6 µm sensor, (**b**) 10 µm sensor, and (**c**) 50 µm sensor for a SARS-CoV-2 RT-LAMP reaction in a Bode plot, which shows impedance (top) and phase (bottom) versus frequency. Impedance spectra are plotted from the start of the reaction 0 min (brown) until 35 min (green) in steps of 5 min. Impedance is decreasing with time. Current measurement by the impedance analyzer undergoes a range switch at around 30 kHz, which results in a buckling curve. Negative resistance contribution in frequencies higher than 30 kHz can be explained by a third electrode (heating meander).

**Figure 8 biosensors-12-00261-f008:**
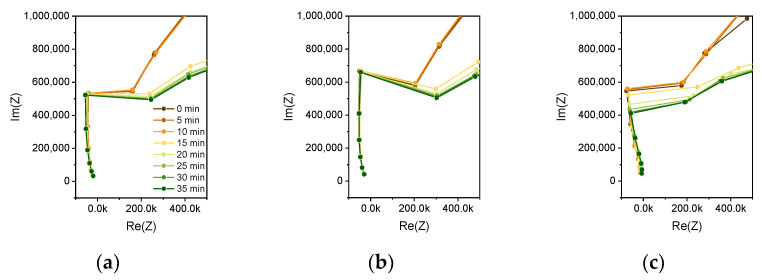
Nyquist plots of (**a**) the 6 µm sensor, (**b**) 10 µm sensor, and (**c**) 50 µm sensor measurement of a SARS-CoV-2 RT-LAMP reaction. Impedance spectra are plotted in steps of 5 min from the start of the reaction 0 min (brown) until 35 min (green). The change in the impedance is mainly happening in the imaginary part of the impedance. Assuming the sensor to be modeled by a parallel RC circuit, the capacitive contribution to the impedance change is therefore dominant. Changes in the dielectric constant are measured by this impedance analysis using interdigitated electrodes.

**Figure 9 biosensors-12-00261-f009:**
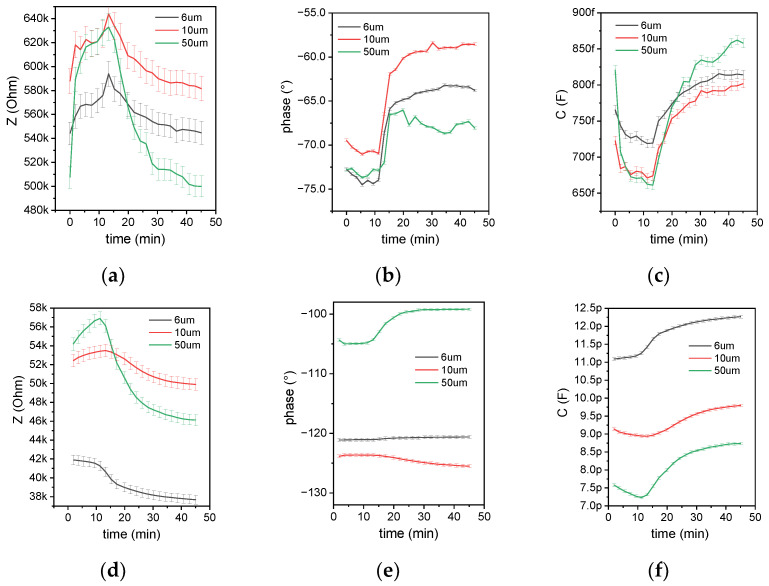
Real-time monitoring of RT-LAMPs targeting SARS-CoV-2 reference RNA and time-dependent data collection of (**a**) impedance, (**b**) phase, and (**c**) capacitance at 25 kHz and (**d**) impedance, (**e**) phase, and (**f**) capacitance at 400 kHz versus time. In the value sequence of the results, three phases can be identified: an initial constant value, an exponential change, and a saturation. The same behavior is observed for qPCR measurements. For small finger distance (6 µm and 10 µm sensor) and high frequency (400 kHz), the signal change is purely capacitive (phase constant in (**e**)) and measurable by the used equipment (impedance < 100 kΩ, C > 9 pF).

**Figure 10 biosensors-12-00261-f010:**
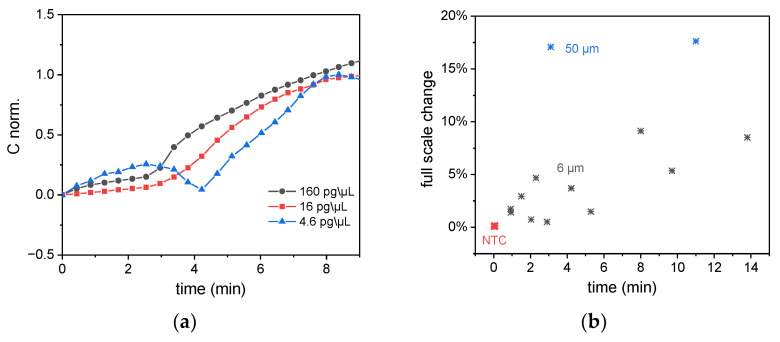
(**a**) Monitoring of SARS-CoV-2 RT-LAMP reactions deploying different target concentrations. The capacitance data has been normalized to improve comparison of the rise time. For less concentrated samples (lower viral copy number), the LAMP reaction starts at a later point in time. (**b**) Distribution of results (x-axis: rise time, y-axis: full-scale change of the capacitance) for the performed amplification experiments on impedance sensors.

**Figure 11 biosensors-12-00261-f011:**
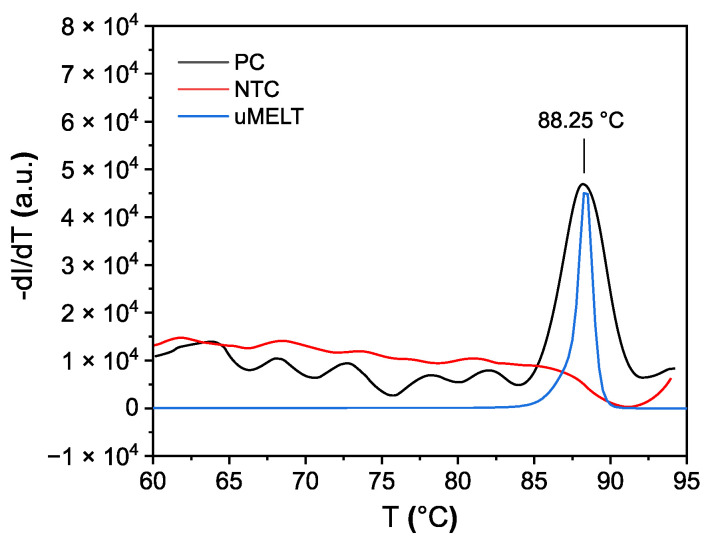
Melt curve analysis of a SARS-CoV-2 RT-LAMP product executed by a qPCR cycler (positive control, PC; nontemplate control, NTC) compared to the theoretically predicted melt curve by uMELT.

**Figure 12 biosensors-12-00261-f012:**
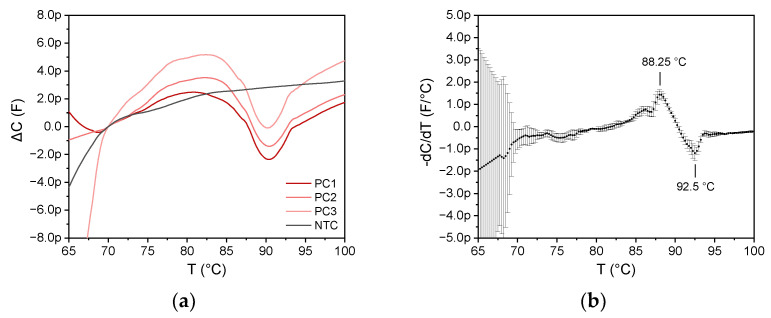
(**a**) Capacitance change measured for parallel experiments of three SARS-CoV-2 LAMP amplicon positive controls (PC, red) and one negative template control (NTC, black) using the 6 µm sensor and the impedance analyzer at 400 kHz. As the sensors exhibit different base capacitance values, data are shown by the change in C referred to as the capacitance value at 70°C. (**b**) Mean value of the derivation of the 3 PC curves in (**a**) with the standard deviation (error bars). The experiment exhibits good reproducibility between 70 and 100°C. The capacitance decrease, which corresponds to denaturation, is identical to the expected theoretical *T_m_* value of 88.25 °C (see Figure 11).

**Figure 13 biosensors-12-00261-f013:**
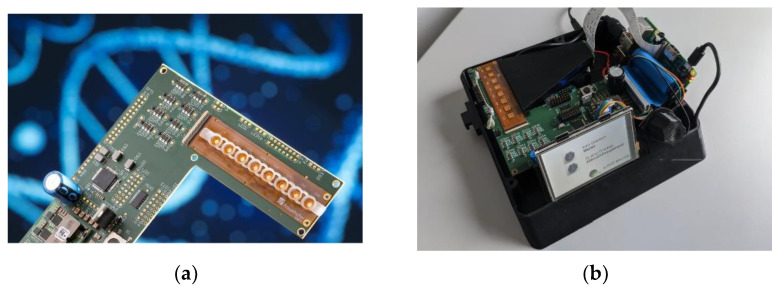
The developed and fabricated demonstrator features (**a**) PCB with the components of the prototype system (see Figure 6a) and the sensor FLEX-PCB. It is part of (**b**) a housing unit with a display and a rotary encoder to guide the user through the steps of sample preparation and extract the results.

**Figure 14 biosensors-12-00261-f014:**
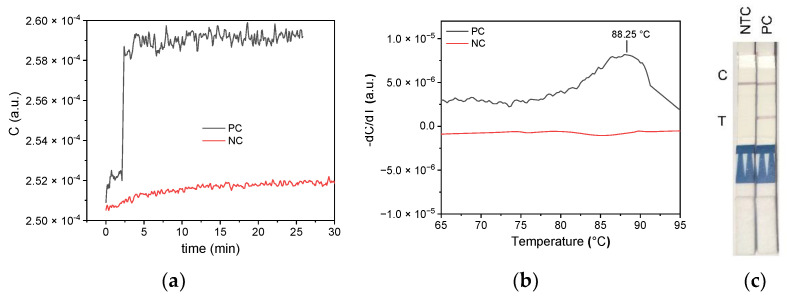
(**a**) LAMP reaction and (**b**) MCA on a 6 µm sensor of 1/10 dilutions of crude SARS-CoV-2-positive Alpha VOC sample EQA 417010 (BetaCoV/Passau/ChVir21652/2020, black) and EQA virus-free sample 409040 (NC, crude cell lysate, red) measured by the touch-sensing controller of the STM32L476 on the demonstrator PCB. Unlike observed for the 417010 positive control, the negative reference sample 409040 does not feature any capacitance changes. (**c**) Final verification of LAMP and MCA results by lateral-flow dipstick analysis. C = internal dipstick control line, T = SARS-CoV-2 positive test line.

**Table 1 biosensors-12-00261-t001:** All SARS-CoV-2-related sequences used in this study.

Designation	Sequence	Genomic Coordinates: (Bases from)
SARS-CoV-2 (GenBank ID: MN908947.3)	GACCCCAAAATCAGCGAAATGCACCCCGCATTACGTTTGGTGGACCCTCAGATTCAACTGGCAGTAACCAGAATGGAGAACGCAGTGGGGCGCGATCAAAACAACGTCGGCCCCAAGGTTTACCCAATAATACTGCGTCTTGGTTCACCGCTCTCACTCAACATGGCAAGGAAGACCTTAAATTCCCTCGAGGACAAGGCGTTCCAATTAACACCAATAGCAGTCCAGATGACCAAATTGGCTACTACCGAAGAGCTACCAGACGAATTCGTGGTGGTGACGGTAAAATGAAAGATCTCAGTCCAAGATGGTATTTCTACTACCTAGGAACTGGGCCAGAAGCTGGACTTCCCTATGGTGCTAACAAAGACGGCATCATATGGGTTGCAACTGAGGGAGCCTTGAATACACCAAAAGATCACATTGGCACCCGCAATCCTGCTAACAATGCTGCAATCGTGCTACAACTTCCTCAAGGAACAACATTGCCAAAAGGCTTCTACGCAGAAGGGAGCAGAGGCGGCAGTCAAGCCTCTTCTCGTTCCTCATCACGTAGTCGCAACAGTTCAAGAAATTCAACTCCAGGCGCAGTAGGGGAACTTCTCCTGCTAGAATGGCTGGCAATGGCGGTGATGCTGCTCTTGCTTTGCTGCTGCTTGACAGATTGAACCAGCTTGAGAGCAAAATGTCTGGTAAAGGCCAACAACAACAAGGCCAAACTGTCACTAAGAAATCTGCTGCTGAGGCTTCTAAGAAGCCTCGGCAAAAACGTACTGCCACTAAAGCATACAATGTAACACAAGCTTTCGGCAGACGTGGTCCAGAACAAACCCAAGGAAATTTTGGGGACCAGGAACTAATCAGACAAGGAACTGATTACAAACATTGGCCGCAAATTGCACAATTTGCCCCCAGCGCTTCAGCGTTCTTCGGAATGTCGCGC	28287 to 29230, (N gene)
SARS-CoV2-1.F3	ATGACCAAATTGGCTACTAC	28515–28534
SARS-CoV2-1.F2	TTCGTGGTGGTGACGG	28554–28569
SARS-CoV2-1.FL	FAM-CCATCTTGGACTGAG	28597–28583
SARS-CoV2-1.F1c	AGCTTCTGGCCCAGTTCC	28630–28613
SARS-CoV2-3.B1c	CAAAGACGGCATCATATGGG	28651–28670
SARS-CoV2-3.BL	ACTGAGGGAGCCTTGA	28676–28691
SARS-CoV2-3.B2	GGATTGCGGGTGCCAATGTG	28726–28706
SARS-CoV2-3.B3	AGCACGATTGCAGCAT	28749–28734
SARS-CoV2-1.FIP	Biotin-AGCTTCTGGCCCAGTTCCTTCGTGGTGGTGACGG	28630–28613; 28554–28569
SARS-CoV2-3.BIP	CAAAGACGGCATCATATGGGGGATTGCGGGTGCCAATGTG	28651–28670; 28726–28706

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
