# Peer review of "Passivated Impedimetric Sensors for Immobilization-Free Pathogen Detection by Isothermal Amplification and Melt Curve Analysis"

_biosensors, 2022, doi:10.3390/bios12050261_

Round 1
Reviewer 1 Report
In this manuscript, authors present an embedded uTAS system for pathogen detection by utilizing isothermal amplification and melt curve analysis with impedance measurements. In the current pandemic context, the presented works are novel and could be very helpful for various purposes. The presented work is well designed and was performed with decent manner. The final deliverable is already a readily system, and the writing is clear and fluent. Overall, the manuscript is in perfect shape. I think the manuscript could be suitable for publication in Biosensor after addressing a few minor concerns below.
- Some of the experiments could be better testified with additional control data. For example, the impedance measurement curve of a blank NTC vs. temperature change is missing; therefore, without a direct comparison it is hard to conclude the proposed system is with good specificity.
- Figure 7,8: any replicates for this particular experiment?
- A limit of detection is typically desired to be discussed, especially in the POC context. Do you have any data to elaborate?
- Line 244: here the capacitance change sensitivity is declared to be 10fF. Is this a typo? If not, 6um should provide a very decent detection power? Please educate.
- Overall, the manuscript has a lot of misarrangements, some of the words are displayed at wrong place. Please have a proofread and revise it accordingly. For example, “Our” in line 108 should be in line 113. Line 108, figure caption shows in the text. Etc.
- Figure reference in text requires extra attentions. Lot of the figure references is wrong or mis-formatted.
- Figure 1 could be more readable with an icon labeling.
- Figure 10b: color coding the 6um and 50um group could help to clarify which data point belongs to which group.
Author Response
Comments and Suggestions for Authors
In this manuscript, authors present an embedded uTAS system for pathogen detection by utilizing isothermal amplification and melt curve analysis with impedance measurements. In the current pandemic context, the presented works are novel and could be very helpful for various purposes. The presented work is well designed and was performed with decent manner. The final deliverable is already a readily system, and the writing is clear and fluent. Overall, the manuscript is in perfect shape. I think the manuscript could be suitable for publication in Biosensor after addressing a few minor concerns below.
I would like to thank the reviewer 1 for accepting our manuscript publication in Biosensor after addressing a few minor concerns below by giving clarifications and answers
- Some of the experiments could be better testified with additional control data. For example, the impedance measurement curve of a blank NTC vs. temperature change is missing; therefore, without a direct comparison it is hard to conclude the proposed system is with good specificity.
Answer: The suggestion of adding additional control data, such as impedance measurement curve of a blank NTC vs. temperature change was taken into account and added to figure 12 and 14.
- Figure 7,8: any replicates for this particular experiment?
Answer: The data already show 3 experiments on three parallel sensors with different electrode gap a. As it is important for the elaboration of the impedance measurement (optimum gap a and optimum measurement frequency) no further experiments were executed.
- A limit of detection is typically desired to be discussed, especially in the POC context. Do you have any data to elaborate?
Answer: Unfortunately, we do not have any additional data at the moment. The LOD can be affected mainly by the set of primers. Your remark points to a n important parameter, which we will focus on in our ongoing research.
- Line 244: here the capacitance change sensitivity is declared to be 10fF. Is this a typo? If not, 6um should provide a very decent detection power? Please educate.
Answer: We want to emphasize that the sensor has to be designed accordingly to the requirements of the TSC. 6 µm is required as it exhibits a C > 10 pF. The good sensitivity of the TSC can still measure the small full-scale change of the 6 µm sensor.
- Overall, the manuscript has a lot of misarrangements, some of the words are displayed at wrong place. Please have a proofread and revise it accordingly. For example, “Our” in line 108 should be in line 113. Line 108, figure caption shows in the text. Etc.
Answer: The manuscript is properly arranged and is now in good shape.
- Figure reference in text requires extra attentions. Lot of the figure references is wrong or mis-formatted.
Answer: Figure references is now arranged.
- Figure 1 could be more readable with an icon labeling.
Answer: We have added labeling to the Figure 1.
- Figure 10b: color coding the 6um and 50um group could help to clarify which data point belongs to which group.
Answer: Color coding was added as well as the NTC value.
Reviewer 2 Report
Very interesting impedimetric sensor for nucleic acids corresponding to Covid-19, including temperature control interface suitable for generating melting curve data after LAMP amplification steps.
Fig. 3 any indication of dimensions, electrodes thicknesses and spacings?
Fig. 4b, meaning of a on the x-axis? Perhaps the a parameter can be indicated in Figure 3? circles and bars correspond to which y-axis - indicate with arrows for clarity.
The impedance sensing principle using the touch detection capability of the Nucleo board is quite smart, though not original - [27,28].
Fig. 7, use the measured points instead of lines, as the sharp changes on the curves look rather strange. Can the disturbing range switch of the instrument be prevented somehow?
Fig. 10a blank experiments should be provided, too (in the absence of the target sequence, i.e. zero concentration). Instead of the "accidental" evaporation, a correct curve without this disturbance should be provided.
L429 - Fig. 18?
Fig. 13, provide block scheme; is Raspberry Pi able to measure touch capacitance changes?
Demonstration of successful performance with real samples containing Covid-19 virus should be provided to confirm the data generated on artificially prepared oligonucleotides.
Author Response
Comments and Suggestions for Authors
Very interesting impedimetric sensor for nucleic acids corresponding to Covid-19, including temperature control interface suitable for generating melting curve data after LAMP amplification steps.
we would like to thank the reviewer 2 for accepting to review our manuscript. We will address the reviewer’s comments below.
Fig. 3 any indication of dimensions, electrodes thicknesses and spacings?
Answer: We have added to figure 3, the dimension corresponding to 1 mm.
Fig. 4b, meaning of a on the x-axis? Perhaps a parameter can be indicated in Figure 3? circles and bars correspond to which y-axis - indicate with arrows for clarity.
Answer: This was added in Figure 3 and colored axis description was added.
The impedance sensing principle using the touch detection capability of the Nucleo board is quite smart, though not original - [27,28].
Answer: The description of the measurement principle is now deleted.
Fig. 7, use the measured points instead of lines, as the sharp changes on the curves look rather strange. Can the disturbing range switch of the instrument be prevented somehow?
Answer: Unfortunately, all device specific enhancements (averaging, measurement time) were already enabled.
Fig. 10a blank experiments should be provided, too (in the absence of the target sequence, i.e. zero concentration). Instead of the "accidental" evaporation, a correct curve without this disturbance should be provided.
Answer: NTC vs. temperature was added to figure 12 and 14. Figure 10a was zoomed in to the important data interval
L429 - Fig. 18?
Answer: corrected
Fig. 13, provide block scheme; is Raspberry Pi able to measure touch capacitance changes?
Answer: Block scheme is provided in Fig 5a. The text is now clearer, that the Raspberry Pi only provides user guidance on the display.
Demonstration of successful performance with real samples containing Covid-19 virus should be provided to confirm the data generated on artificially prepared oligonucleotides.
Answer: Additional experiments were conducted with sample from ring trials and added to Figure 14.
Round 2
Reviewer 1 Report
In good shape, and has addressed all comments in the previous review. I would like to recommend it to be published.
Reviewer 2 Report
The suggested improvements and corrections were properly realized, now the paper is suitable for publication.